# Trajectories of Hospitalization Cost Among Patients of End-Stage Lung Cancer: A Retrospective Study in China

**DOI:** 10.3390/ijerph15122877

**Published:** 2018-12-15

**Authors:** Zhong Li, Shan Jiang, Ruibo He, Yihan Dong, Zijin Pan, Chengzhong Xu, Fangfang Lu, Pei Zhang, Liang Zhang

**Affiliations:** 1School of Medicine and Health Management, Huazhong University of Science and Technology, Wuhan 430030, China; lizhong@hust.edu.cn (Z.L.); heruibo27@163.com (R.H.); ZJSXdong_yihan@163.com (Y.D.); m201775315@hust.edu.cn (Z.P.); 2School of Health Policy and Management, Nanjing Medical University, Nanjing 211166, China; 15720801899@163.com; 3Yichang Center for Disease Control and Prevention, Yichang 443000, China; function68@hotmail.com (C.X.); fangfanglu91@126.com (F.L.); 13477841441@163.com (P.Z.)

**Keywords:** end-of-life, lung cancer, cost trajectory, place of death, palliative care, latent class analysis, China

## Abstract

This study was conducted to investigate the trajectory of hospitalization costs, and to assess the determinants related to the membership of the identified trajectories, with the view of recommending future research directions. A retrospective study was performed in urban Yichang, China, where a total of 134 end-stage lung cancer patients were selected. The latent class analysis (LCA) model was used to investigate the heterogeneity in the trajectory of hospitalization cost amongst the different groups that were identified. A multi-nominal logit model was applied to explore the attributes of different classes. Three classes were defined as follows: Class 1 represented the trajectory with minimal cost, which had increased over the last two months. Classes 2 and 3 consisted of patients that incurred high costs, which had declined with the impending death of the patient. Patients in class 3 had a higher average cost than those in Class 2. The level of education, hospitalization, and place of death, were the attributes of membership to the different classes. LCA was useful in quantifying heterogeneity amongst the patients. The results showed the attributes were embedded in hospitalization cost trajectories. These findings are applicable to early identification and intervention in palliative care. Future studies should focus on the validation of the proposed model in clinical settings, as well as to identify the determinants of early discharge or aggressive care.

## 1. Introduction

The end-of-life (EOL) costs for cancer patients are of global concern [1,2,3]. Approximately 10% of the overall lifetime healthcare consumption occurs within the final 12 months of a patient’s life [4], during which time decedents with cancer undergo intensive medical treatment before death [5]. Complicated symptoms force end-stage cancer patients to seek care from the emergency department during their last years of life [6]. However, many people cannot afford the potentially life-saving or life-prolonging treatments that are available from hospitals, and this phenomenon is even worse in developing countries [7]. Aggressive care is the main driver of EOL costs, and it places substantial pressure on the consumption of scarce healthcare resources [8]. It was revealed that patients with intensive healthcare utilization underwent resource abuse during their final days of life [9]. In addition, whilst most palliative care services are provided in developed countries, nearly 80% of the global need is situated in developing countries; thus, the lack of access to palliative care is a global concern [10,11]. The World Bank income group, and the availability of national palliative care funding are also heavily correlated [12]. WHO has recommended that national palliative care systems be provided with adequate resources [13], and the World Health Assembly (WHA) 67.19 in 2014 pointed out that palliative care should be integrated as a part of the healthcare framework throughout a patient’s life [14]. In the USA, a hospice programme provides services for patients with a maximum expected survival period of six months [15]. EOL care in Canada includes inpatient hospitalization, chronic care facilities, or at-home care [16]. A study using health administrative data revealed that in the USA and Ontario, the mean monthly cost of a lung cancer patient increased with the patient’s impending death, mainly driven by inpatient hospitalization [17]. In Europe, many patients with end-stage diseases preferred a home death, but most of the patients died in hospitals [18,19]. Cancer specialists did not practice early palliative care as expected [20], and given the benefits of palliative care for the society, especially for the poor [21,22,23] and in developing countries, timely intervention in patients with incurable diseases has thus been widely proposed [24,25].

As the leading type of cancer, lung cancer accounts for 11.6% of the total cancer incidence, and 18.4% of the global mortality [26]. China recorded the highest incidence of death due to lung cancer, accounting for nearly 40% of global cancer-related deaths [12]. The Chinese government and medical institutions have widely promoted palliative medicine and hospice care [27]. However, with scarce resources, oncologists in tertiary hospitals struggle to meet the needs of cancer patients. Occasionally, the oncologists must persuade their patients to receive supportive care in local healthcare facilities or at home. Consequently, the practitioners at the primary care facilities were unable to meet the patients’ needs, including the palliation of symptoms and the management of pain [28]. Some patients and families must wait to be admitted to tertiary hospitals, or they face death [29,30]. However, the timing and the means of providing palliative care for patients with incurable and terminal illnesses in China are unclear and difficult to explore. Therefore, cross-national comparisons of EOL care based on different metrics are necessary [9].

## 2. Methods

### 2.1. Data Source and Study Population 

Based on the International Statistical Classification of Diseases and Related Health Problems 10th Revision, specifically, the WHO version for 2016 [31], 894 decedents diagnosed with cancer between July 2015 and June 2017 in urban Yichang, China, were enrolled into the study. The demographics of the cancer decedents, including place of death, type of cancer, and date of death, were collected from the National Population Death Registration and Management System. The Yichang Health Management Centre, which is affiliated with the Yichang Centre for Diseases Control and Prevention, has integrated hospital information systems, a health insurance database, and a population information database with patients’ identification card numbers. This study was approved by the ethics committee of Tongji Medical College, Huazhong University of Science and Technology (IORG No: 2018S291). All of the data were de-identified. 

### 2.2. Study Design

Previous studies have faced groups of patients with different unobservable preferences and needs. These studies did not identify the actual differences based on the treatment pathways [32], but rather, the studies focused solely on the description of the differences in frequencies, thereby lowering the effects of timely intervention [33]. In 1968, Lazarsfeld and Henry introduced the latent class analysis (LCA) model to identify latent categorical attitude variables [34]. This method was widely used in marketing studies to classify consumers into specific segments to maximize within-segment homogeneity and between-segment heterogeneity, based on the response patterns [35]. In the field of health service science, LCA was applied to examine unobserved classes in healthcare utilization and cost [36,37]. Louviere argued for the frequent use of the LCA model, owing to its convenience in estimation and interpretation when used in health technology assessments [38]. However, diagnosis-based trajectory groups may not reflect the actual trajectory shapes within specific diagnostic groups, due to functional decline or healthcare utilization [39]. Several studies have shown that identifying the cost trajectory could help in implementing timely palliative care [5,40,41]. Fassbender et al. [40] attempted to unify death trajectories with cost data. Lunney et al. [42,43] evaluated the patterns of EOL healthcare expenditures using diagnosis-based trajectory groups. Several methods were applied to investigate the death trajectories and healthcare utilization costs of specific diseases, especially the EOL cost, based on the different kinds of databases [44,45], thereby indicating the importance of such trajectories in palliative care [46]. In this study, a total of 310 decedents diagnosed with lung cancer were initially included, and then 134 decedents who had survived for at least six months were the final participants included in the study. LCA, an alternative method to identify heterogeneity amongst patients, was performed on the enrolled patients to: (1) group the decedents with lung cancer based on the EOL hospitalization cost; (2) explore the attributes of the identified classes; and (3) make recommendations for future directions.

### 2.3. Statistical Analysis

First, descriptive analysis was used to enable a detailed description of the characteristics of the enrolled decedents. Second, LCA is a widely used data analysis approach to identify unobserved heterogeneity in a specific population [47,48]. Monte Carlo simulation has proven that the Bayesian Information Criteria (BIC) is the best amongst the information criteria for homogeneous class enumeration [49]. Moreover, model selection should include the theoretical meanings (i.e., parsimony and interpretability) [31,50]. In this study, zero hospitalization cost was frequent, and cost frequency reduced the increase in cost. Thus, the censored normal model with the traj procedure was applied for data modelling [51]. Third, the Shapiro–Wilks test of normality was performed, as well as the Chi-squared test (Fisher exact test, if necessary), Analysis of Variance, and the Kruskal–Wallis test, followed by the least significance difference and Dunn–Bonferroni post hoc method, which were used to compare the differences between the different classes. Finally, the multi-nominal logit model was used to identify the determinants of different cost categories [52]. The independent variables were as follows: (1) age, (2) gender, (3) level of education, (4) marital status, (5) type of medical insurance, (6) number of emergency department visits, (7) number of ICU admissions, (8) number of inpatients, (9) number of outpatients, and (10) period of survival. All the above-mentioned procedures were calculated using Stata 14.0. The statistical significance level was *p* < 0.05. The level of statistical marginal significance was set at 0.05 < *p* < 0.10, as described in [53].

## 3. Results

### 3.1. Demographic Characteristics

As shown in Table 1, amongst the 134 decedents, 74.6% (100) were male. The median age was 70 years old (p25–75, 63–78), and 85.8% (115) lived with their spouse. Approximately 71.6% (96) were enrolled in the Urban Employee Basic Medical Insurance scheme (UEBMI), and 73.1% (98) had a diploma or had only achieved a junior school level of education. The median survival period was 350 days (p25–75, 242–490), and 33.6% of the patients died at home (Table 1).

### 3.2. Latent Class Membership

Table 2 shows the results of model fitting, including the number of classes, the polynomial order of coefficients, the BIC, Akaike Information Criterion, and the log Bayes factor. The three-class model was the model that was best-suited to describe the heterogeneity of hospitalization cost trajectories, based on the comparison of fit indices with smaller BIC values than those of other models. Hence, the 134 decedents were grouped into three classes, namely, persistently low and increasing (Class 1), persistently high and decreasing (Class 2), and moderate and decreasing (Class 3), as shown in Figure 1.

As shown in Table 1, Class 1 (25.4%) consisted of decedents who were discharged from the hospital early prior to their death, and who were gradually readmitted to the hospital in their last two months of life, with an extremely low hospitalization cost. Class 2 (50.0%) contained decedents with a moderate cost of hospitalization that showed a decreasing trend in cost. In Class 3 (24.6%), the patients had an extremely high cost with a decreasing trend and a high intensity of care, where nearly all of the patients (97%) died in the hospital.

The demographic and economic characteristics of each class were examined. No significant differences in age (*p* = 0.444), gender (*p* = 0.095), survival period (*p* = 0.724), and outpatient services (*p* = 0.185) were observed amongst the three classes. Regarding marital status, the decedents in Class 1 exhibited a lower marriage rate (71%) than those in Classes 2 (91%, Chi-squared = 7.08, *p* = 0.008) and 3 (91%, Chi-squared = 4.42, *p* = 0.035). Class 3 had a higher rate of UEBMI enrolment (91%), compared to Classes 1 (91%, Chi-squared = 9.58, *p* = 0.008) and 2 (91%, Chi-squared = 8.15, *p* = 0.017). Class 3 had a higher level of education (i.e., reached at least senior school level) than Class 2 (81%, Chi-squared = 7.82, *p* = 0.005). For POD, Class 3 had a higher rate (97%) of death in medical institutions than Classes 1 (91%, Chi-squared = 22.34, *p* < 0.001) and 2 (91%, Chi-squared = 13.51, *p* < 0.001). Regarding hospitalization frequency and EOL hospitalization cost, Class 3 had the highest hospitalization frequency, followed by Classes 2 and 1. Class 3 also had a higher median occurrence of emergency department visits than class 1 (Chi-squared = 6.43, *p* = 0.011).

### 3.3. Class Membership Determinants

In this study, the *R*^2^ generated from the multi-nominal logistic regression model was 39.69%, indicating an adequate model fitness as described in Reference [54]. The different classes had three attributes. As shown in Table 3, although some *p*-values were marginally significant, individuals who had received at least a senior school diploma had a 4.71-fold (RR = 4.71, *p* = 0.093) and 3.48-fold (RR = 3.48, *p* = 0.039) probability of being classified in Class 3 than in Classes 1 and 2, respectively. For individuals who died at home, the probability of being classified in Classes 2 and 3 decreased by 78.2% (RR = 0.258, *p* = 0. 064) and 98.4% (RR = 0.0160, *p* = 0.002), respectively, compared to Class 1. The possibility of being classified in Class 3 decreased by 99.4% (RR = 0.062, *p* = 0.017) compared to Class 2. When the occurrence of hospitalization increased by one unit, individuals had a 3.87-fold (RR = 3.87, *p* < 0.001) and 5.69-fold (RR = 5.69, *p* < 0.001) probability of being classified in Classes 2 and 3, respectively, compared to Class 1. Individuals had a 1.47-fold probability (RR = 1.47, *p* < 0.001) of being classified in Class 3 than in class 2.

## 4. Discussion

LCA is widely applied in cancer-related studies of clinical epidemiology and health economics [5,39,55,56,57]. Real-world data is ideal when investigating the EOL care cost in a large population [58]. To the best of our knowledge, this study was the first to apply LCA to the quantification of hospitalization cost trajectories amongst decedents with end-stage lung cancer in China. This study made a substantial contribution to the current research on cancer cost trajectory, especially its determinants. These results have potential use in the timely intervention of palliative care.

Three classes were identified throughout the evolution of hospitalization cost, based on the BIC and the interpretability of the model. This result was consistent with a study conducted in 2015 using a representative sample of Swiss decedents [5]. However, one class (i.e., 1% of patients aged 66 years old and 4% of young individuals) in the Swiss decedents study showed a steep increase in the average monthly healthcare expenditure in their last month of life [5], indicating the use of intensive medical treatment prior to death. The overall survival period from first-diagnosis to death amongst the decedents enrolled was 350 days. Approximately 85.8% of the patients lived with a spouse. The POD was a valid indicator of where care was provided in the final days or hours of life [59]. The overall rate of home death was 33.6%, thereby ranking at a medium level when compared to the home death rates in 14 countries during 2008 [12]. The above results indicated the existence of a certain level of EOL care intensity. A sizable proportion of decedents still utilized healthcare resources [60], thereby hampering the promotion of home-based palliative care and home death.

The trajectories of Class 2 and 3 showed a considerable decline over the last six months of life when compared to Class 1. The Class 1 curve remained stable, and increased with impending death. This result may be due to the disregard of aggressive care by some patients when death was expected, resulting from the patient’s acceptance and wish for a good death [61]. However, the patients in Classes 2 and 3 and their caregivers did not seem to have fully discussed their approaching death at an early stage, resulting in a high rate of inpatient hospitalization and hospital death. This result was also consistent with the results of a comparative study between the USA and Ontario, Canada [62]. Moreover, the availability of hospice care or willingness to receive palliative care may also explain such differences [8]. Several studies have shown that most patients with advanced cancer faced an increase in symptom burden during the EOL period, thereby forcing them to seek care in hospitals [8]. Management of anxiety, confusion, or delirium in a home setting can be very difficult. Thus, an emergency department visit or hospitalization may be precipitated by the patients’ families, due to distress [63,64]. For the patients in Classes 2 and 3, especially the Class 3 patients with a high hospitalization cost, the high cost may have been because the patients were not informed of the approaching death [65]. The services that these patients received were to prolong their lives instead of the palliation of their symptoms [66]. In addition, the Class 3 patients accounted for 24.6% of the enrolled patients. However, the overall survival period of these patients was not higher than that of Class 1 and 2 patients. This finding was similar to the evidence found in the USA, where high-intensity care was not related to improved outcomes [67].

As described previously, the traditional regression-based method for exploring the determinants of EOL cost neglected the heterogeneity that was formed by the treatment intensity amongst the patients [39,59]. The predictors, namely, level of education, POD, and inpatient hospitalization, for the three classes, were identified by using the multi-nominal logit regression model. Amongst the three classes, the highest rates of being married, of death in hospitals (97%), and the highest amount of hospitalization services were found in Class 3 patients. Patients with a low education level possessed an increased risk of poor healthcare utilization, which may have been influenced by multichannel factors, such as resource availability and financial capability of their families [68]. Hence, enhancement of consultation services should be urgently established to improve EOL palliative care [61]. In this study, home death decreased the probability of high-cost trajectories (i.e., Classes 2 and 3). The higher the frequency of inpatient hospitalization of the individual, the higher the probability of being grouped in Class 2 and 3. Hospital death and inpatient hospitalization could increase the EOL healthcare cost [9,12,14,17]. Shepperd et al. pointed out that early assisted discharge could reduce the hospitalization cost [69]. Furthermore, home hospice care was a cost-saving alternative to inpatient hospice care in many countries [58].

In general, our findings described capacity-building opportunities for a quality palliative care system. Failure to identify the need for palliative care, and intervention of early discharge for the decedents caused the above phenomenon. Based on these results, we can cautiously speculate on the presence of a huge potential for cost-saving that is associated with the improvement of the cancer healthcare delivery system. Hence, treatment variation and heterogeneity should be actively addressed. As family physicians play an important role in hospice care in many countries [12,17], substantial challenges exist for the Chinese primary care system that should also be addressed [70]. Providing timely intervention in targeted palliative care should also be addressed together with the families [12].

Similar to other research, this study had limitations, which included the sample size and the presence of few marginally significant *p*-values. Large-scale studies using real-world data should be conducted. Moreover, preference-related, family, and community-level variables should also be collected. Future works should focus on: (1) The validation of the model based on additional risk or protective factors and the usefulness of the model for clinical decision-making, and (2) The identification of potential determinants of early discharge and aggressive care in end-stage patients. 

## 5. Conclusions

This study was the first to explore the trajectory of hospitalization costs amongst patients with end-stage lung cancer. Different classes of patients had different preferences in healthcare utilization and cost. The results of this study will help to improve the palliative care system, including targeted information and early discharge from hospitals when possible. Such results will also help physicians, policymakers, patients, and their families in making shared decisions. 

## Figures and Tables

**Figure 1 ijerph-15-02877-f001:**
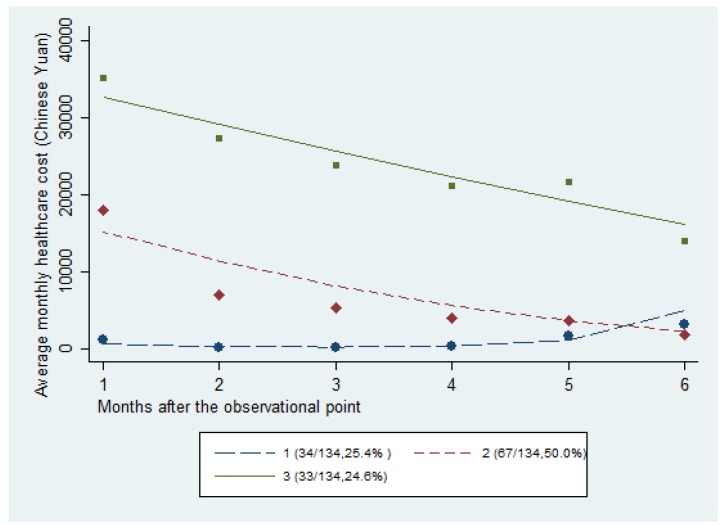
Results of the latent class analysis model.

**Table 1 ijerph-15-02877-t001:** Comparison of demographic characteristic, EOL care utilization, and hospitalization cost.

Characteristic	Overall	Class 1 (n = 34)	Class 2 (n = 67)	Class 3 (n = 33)	$\chi^{2}/F$	*P*-Value
**Age (n = 134), p50 (p25–75)**	70 (63–78)	76 (62–80)	69 (63–78)	70 (67–75)	1.626	0.444
Gender						
Male	100 (74.6%)	23(68%)	48 (72%)	29 (88%)	4.708	0.095
Female	34 (25.4%)	11(32%)	19 (28%)	4 (12%)		
**Marital status**						
Married	115 (85.8%)	24 (71%)	61 (91%)	30 (91%)	7.697	**0.021**
Others	19 (14.2%)	10 (29%)	6 (9%)	3 (9%)		
**Insurance type**						
UEBMI	96 (71.6%)	21 (62%)	45 (67%)	30 (91%)	14.064	**0.007**
URBMI	19 (14.2%)	6 (18%)	10 (15%)	3 (9%)		
NRCMS	19 (14.2%)	7 (20%)	12 (18%)	0		
**Education**						
<= Junior school (n = 98)	98 (73.1%)	25(74%)	54 (81%)	19 (58%)	10.608	**0.031**
>= Senior school (n = 36)	36 (26.9%)	9 (26%)	13 (19%)	14 (42%)		
**Place of death**						
Medical institution	89 (66.4%)	15 (44%)	42 (63%)	32 (97%)	26.899	**<0.001**
Home	45 (33.6%)	19 (56%)	25 (37%)	1 (3%)		
Survival (Days)	350 (242–490)	347 (225–521)	329 (248–443)	363 (231–528)	0.645	0.724
OP	4 (1–12)	3 (1–10)	7 (2–14)	3 (1–12)	3.381	0.185
EDV	3 (1–5)	1 (1–2)	3 (2–5)	5 (3–6)	51.988	**<0.001**
IHS #	1 (0–3)	1 (0–2)	1 (0–3)	2 (1–4)	6.092	**0.048**
ICU	0 (0–0)	0 (0–0)	0 (0–0)	0 (0–0)	1.526	0.466
Hospitalization cost in the last six months *	30996 (11504–85896)	445 (0–7108)	31141 (18582–56415)	128165 (98924–163944)	102.389	**<0.001**

Note: EOL, End-of-life; UEBMI, Urban Employee Basic Medical Insurance; URBMI, Urban Resident Basic Medical Insurance; NRCMS, New Rural Cooperative Medical System; OP, outpatient services; IHS, inpatient hospitalization services; EDV, emergency department visit; ICU, intensive care unit. # 3 > 2 (Chi-squared = 7.67, *p* = 0.006), 2 > 1 (Chi-squared = 35.97, *p* < 0.001), 3 > 1 (Chi-squared = 41.46, *p* < 0.001); * 3 > 2 (Chi-squared = 60.90, *p* < 0.001), 2>1 (Chi-squared = 51.79, *p* < 0.001), 3 > 1 (Chi-squared = 50.32, *p* < 0.001).

**Table 2 ijerph-15-02877-t002:** Results of model fitness.

Number of Class	Polynomial Order of Coefficients	BIC (N = 134)	AIC (N = 134)	Log Bayes Factors
1	1	−4999.24	−4994.89	4991.89
2	22	−4923.09	−4911.5	4903.5
3	211	−4916.23	−4901.74	4891.74
4	1113	−4925.77	−4905.48	4891.48

Note: BIC, Bayesian Information Criterion; AIC, Akaike Information Criterion.

**Table 3 ijerph-15-02877-t003:** Determinants of the membership in the different classes.

Variables	Groups	Class 2 (vs Class 1)			Class 3 (vs Class 1)			Class 3 (vs Class 2)		
		Beta	*P*	95% CI	Beta	*P*	95% CI	beta	*P*	95% CI
**Gender**	Female (vs Male)	−0.002	0.998	(−1.421–1.417)	−0.845	0.397	(−2.800–1.109)	−0.842	0.261	(−2.313–0.628)
**Age**	65–80 (vs<65)	0.222	0.78	(−1.338–1.782)	0.473	0.639	(−1.501–2.447)	0.251	0.73	(−1.173–1.676)
	>80 (vs<65)	0.329	0.765	(−1.827–2.484)	0.318	0.821	(−2.430–3.065)	−0.01	0.992	(−2.011–1.99)
**Health Insurance**	URBMI (vs UEBMI)	−0.2	0.863	(−2.452–2.054)	15.278	0.995	(−4682–4713)	15.478	0.995	(−4682–4713)
	NRCMS (vs UEBMI)	−0.91	0.36	(−2.848–1.034)	14.567	0.995	(−4683–4712)	15.474	0.995	(−4682–4713)
**Marital status**	Others (vs Married)	0.989	0.26	(−0.731–2.709)	1.288	0.309	(−1.191–3.768)	0.299	0.76	(−1.615–2.214)
**Education**	≥SS (vs≤ JS)	0.303	0.701	(−1.245–1.852)	1.55	0.093	(−0.26–3.361)	1.247	0.039	(0.063–2.431)
**POD**	Home (vs Hospital)	−1.35	0.064	(−2.784–0.079)	−4.137	0.002	(−6.79–1.484)	−2.784	0.017	(−5.073–0.496)
**Survival**		−0.002	0.244	(−0.007–0.002)	−0.003	0.172	(−0.009–0.002)	−0.001	0.53	(−0.005–0.003)
**OP**		0.021	0.543	(−0.046–0.088)	0.016	0.704	(−0.067–0.101)	−0.004	0.873	(−0.061–0.052)
**EDV**		−0.06	0.588	(−0.263–0.149)	−0.037	0.729	(−0.252–0.176)	0.019	0.72	(−0.085–0.124)
**IHS**		1.354	<0.001	(0.749–1.959)	1.739	<0.001	(1.0864–2.392)	0.385	0.005	(0.118–0.652)
**ICU**		0.94	0.42	(−1.344–3.223)	1.744	0.199	(−0.918–4.408)	0.805	0.323	(−0.791–2.401)

Note: In this model, number of observation = 131, LR chi2(26) = 107.83, *p* < 0.001, Pseudo R-squared = 0.3969. UEBMI, the Urban Employee Basic Medical Insurance; URBMI, Urban Resident Basic Medical Insurance; NRCMS, New Rural Cooperative Medical System; POD, place of death; JS, Junior School; SS, Senior School; OP, outpatient services; IHS, inpatient hospitalization services; EDV, emergency department visits; ICU, intensive care unit.

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
