# Peer review of "Trajectories of Hospitalization Cost Among Patients of End-Stage Lung Cancer: A Retrospective Study in China"

_ijerph, 2018, doi:10.3390/ijerph15122877_

Round 1
Reviewer 1 Report
In general, there is a lack of explanation of the selected model on the section of Methods, it need to explain why this method is better or different from other methods. There are several results not supported with data, on section of Results, “73.3% (98) had a diploma or achieved only junior school level of education”, which is inconsistent with the data of 73.1% (98) in Table 1. Furthermore, there are some abbreviations are not shown in the table (POD, JS, SS in Table 1), and there is no explanation for some abbreviations in the table (HIS in Table 3). To be honest, I don't know how to understand “class 4” in Table 2, which is not mentioned in the article.
Author Response
Point 1: there is a lack of explanation of the selected model on the section of Methods, it need to explain why this method is better or different from other methods.
Response 1: Thank you very much for this valuable suggestion, previously, we attempted to describe the explanation of the selected in the Introduction section, given this comments and suggestions from the Reviewer 2 and you, we removed this part into the section of Methods as followed:
2.2. Study Design
Previous studies have faced groups of patients with different unobservable preferences and needs. These studies did not identify the actual differences based on the treatment pathways [32], but rather the studies focused solely on the description of the differences in frequencies, thereby lowering the effects of timely intervention [33]. In 1968, Lazarsfeld and Henry introduced the latent class analysis (LCA) model to identify latent categorical attitude variables [34]. This method was widely used in marketing studies to classify consumers into specific segments to maximize within-segment homogeneity and between-segment heterogeneity based on the response patterns [35]. In the field of health service science, LCA was applied to examine unobserved classes in healthcare utilization and cost [36, 37]. Louviere argued for the frequent use of the LCA model owing to its convenience in estimation and interpretation when used in health technology assessments [38]. However, diagnosis-based trajectory groups may not reflect the actual trajectory shapes within specific diagnostic groups due to functional decline or healthcare utilization [39]. Several studies have shown that identifying the cost trajectory could help in implementing timely palliative care [5, 40, 41]. Fassbender et al. [40] attempted to unify death trajectories with cost data. Lunney et al [42, 43] evaluated the patterns of EOL healthcare expenditures using diagnosis-based trajectory groups. Several methods were applied to investigate the death trajectories and healthcare utilization costs of specific diseases, especially the EOL cost, based on the different kinds of databases [44, 45], thereby indicating the importance of such trajectories in palliative care [46]. In this study, a total of 310 decedents diagnosed with lung cancer were initially included, and then 134 decedents who had survived for at least 6 months were the final participants included in the study. LCA, an alternative method to identify heterogeneity amongst patients, was performed on the enrolled patients to: 1) Group the decedents with lung cancer based on the EOL hospitalization cost; 2) Explore the attributes of the identified classes; and 3) Make recommendations for future directions.
Point 2: There are several results not supported with data, on section of Results, “73.3% (98) had a diploma or achieved only junior school level of education”, which is inconsistent with the data of 73.1% (98) in Table 1.
Response 2: We are so sorry about this mistake, and we have corrected it in the manuscript.
Point 3: there are some abbreviations are not shown in the table (POD, JS, SS in Table 1), and there is no explanation for some abbreviations in the table (HIS in Table 3).
Response 3: We are so sorry about this mistake, and we have added the abbreviations below the tables.
Point 4: To be honest, I don't know how to understand “class 4” in Table 2, which is not mentioned in the article.
Response 4: We are so sorry for this mistake that not describe the results of latent class analysis in details. We performed this method with different number of class, using different order of coefficients. According to the criterial of the BIC value, the study population was divided into 3 classes (Nylund K L, 2007).
Reviewer 2 Report
The study aims to group the decedents of lung cancer into three types (classes) with the latent class analysis of their end-of-life hospitalization cost.
1. Title: The title could be shortened as “Trajectories of hospitalization cost among patients of end-stage lung cancer: a retrospective study in China”.
2. Introduction: The introduction is too lengthy and could be shortened to one or two paragraphs.
3. The aim of the study (Lines 94-97): The third purpose “3) inform the timely intervention of palliative care” can be hardly proved from the study. Instead, it is the implication or future direction.
4. Methods: Please consider dividing the whole section into 2.1. Data source and study population, 2.2. Study design, and 2.3. Statistical analysis.
5. Table 1 – footnotes (Lines 138-139): Some terms are unnecessary (POD, place of death; JS, junior school; SS, Senior school). Some terms are lacking (EMC; ICU).
6. Conclusion (Lines 277-279). Please consider moving the sentences “Similar to other research, this study has limitations, which are the sample size and few marginally significant P-values. Large-scale studies using real-world data should be conducted.” to the end of Discussion section as a separate paragraph with more other limitations.
Author Response
Point 1: Title: The title could be shortened as “Trajectories of hospitalization cost among patients of end-stage lung cancer: a retrospective study in China”. Response 1: Thank you very much for this valuable suggestion, and we have revised the title. Point 2: Introduction: The introduction is too lengthy and could be shortened to one or two paragraphs. Response 2: thank you very much for this question and we have tried to simplify the paragraph, including removing the methodology-related paragraph into the Method section. Point 3: The aim of the study (Lines 94-97): The third purpose “3) inform the timely intervention of palliative care” can be hardly proved from the study. Instead, it is the implication or future direction. Response 3: Thanks so much for this valuable suggestion, we have revised it. Point 4: Methods: Please consider dividing the whole section into 2.1. Data source and study population, 2.2. Study design, and 2.3. Statistical analysis. Response 4: Thank you much for this valuable suggestion, we have revised it as you suggested, and added the brief introduction of latent class analysis in the 2.2. Study design to tell the readers why we choose the statistical technology. Point 5: Table 1 – footnotes (Lines 138-139): Some terms are unnecessary (POD, place of death; JS, junior school; SS, Senior school). Some terms are lacking (EMC; ICU). Response 5: We are so sorry about this mistake, and we have revised it as you suggested. Point 6: Conclusion (Lines 277-279). Please consider moving the sentences “Similar to other research, this study has limitations, which are the sample size and few marginally significant P-values. Large-scale studies using real-world data should be conducted.” to the end of Discussion section as a separate paragraph with more other limitations. Response 6: Thank you so much for this suggestion, and we revised it as followed.
Round 2
Reviewer 2 Report
The section of "Author Contributions" has not been suitably written.